# Large-Scale Field Cultivation of *Morchella* and Relevance of Basic Knowledge for Its Steady Production

**DOI:** 10.3390/jof9080855

**Published:** 2023-08-16

**Authors:** Wei Liu, Peixin He, Xiaofei Shi, Ya Zhang, Jesus Perez-Moreno, Fuqiang Yu

**Affiliations:** 1The Germplasm Bank of Wild Species, Yunnan Key Laboratory for Fungal Diversity and Green Development, Kunming Institute of Botany, Chinese Academy of Sciences, Kunming 650201, China; zhenpingliuwei@mail.kib.ac.cn (W.L.); shixiaofei@mail.kib.ac.cn (X.S.); 2College of Food and Biological Engineering, Zhengzhou University of Light Industry, Zhengzhou 450002, China; peixinhe191@163.com; 3Sichuan Junyinong Agricultural Technology Co., Ltd., Chengdu 610023, China; zhangya654321@126.com; 4Edafologia, Campus Montecillo, Colegio de Postgraduados, Texcoco 56230, Mexico

**Keywords:** morels, cultivation mode, exogenous nutrition bag, feeding technology, spawn, heterothallic life cycle, mating type, strain aging

## Abstract

Morels are one of the most highly prized edible and medicinal mushrooms worldwide. Therefore, historically, there has been a large international interest in their cultivation. Numerous ecological, physiological, genetic, taxonomic, and mycochemical studies have been previously developed. At the beginning of this century, China finally achieved artificial cultivation and started a high-scale commercial development in 2012. Due to its international interest, its cultivation scale and area expanded rapidly in this country. However, along with the massive industrial scale, a number of challenges, including the maintenance of steady economic profits, arise. In order to contribute to the solution of these challenges, formal research studying selection, species recognition, strain aging, mating type structure, life cycle, nutrient metabolism, growth and development, and multi-omics has recently been boosted. This paper focuses on discussing current morel cultivation technologies, the industrial status of cultivation in China, and the relevance of basic biological research, including, e.g., the study of strain characteristics, species breeding, mating type structure, and microbial interactions. The main challenges related to the morel cultivation industry on a large scale are also analyzed. It is expected that this review will promote a steady global development of the morel industry based on permanent and robust basic scientific knowledge.

## 1. Introduction

True morel is the general name of all species of *Morchella* that belong to the ascomycetes, and they are typical representatives of macro-ascomycetes [1,2]. *Morchella* species have high nutritional and medicinal value, with a long consumption culture all over the world, and they constitute a rich source of bioactive compounds that improve human health and well-being [2,3,4,5]. Antidiabetic, anticarcinogenic, anti-inflammatory, antimicrobial, antioxidant, and hepatoprotective properties have been found in morel species [6,7,8,9]. Morels have great global relevance, not only because they are widely appreciated wherever they grow but also because their international commerce is immense. Additionally, cultivated morels have not yet replaced wild-harvested morels. However, official, accurate data on morel global production are lacking. Despite this, there are some data that demonstrate that morels are among the most highly prized mushrooms in the world, e.g., (i) In 2007, it was reported that only in western North America did their annual trade range from USD 5 to 10 million [2]. (ii) From 2010 to 2015, in China, the annual export of dried morels increased fivefold from 181 to 900 tons, averaging USD 160 per kilogram [10]; and (iii) the average current price of USD 200 per kilogram is very common in European or North American markets; and retail prices around the world can reach up to tenfold. As a famous gourmet edible mushroom, its biology and domestication have long been pursued by researchers all over the world. The systematic and comprehensive research of morel biology was boosted in the 1990s, which mainly included studies related to nutrition type, nutrition metabolism, sclerotia development, heterothallism, and the life cycle [11,12,13,14,15,16,17,18,19]. However, due to the inability to complete the complete life cycle of morel under artificial conditions, all the research stagnated at that time, and it was reduced to studies related to taxonomy and nutritious and bioactive compound composition. Due to its complex biology, domestication and cultivation of the morel have historically been difficult challenges worldwide. The internationally recognized indoor cultivation technology at a low scale for morels was developed by R.D. Ower in the USA [12]. This technology mainly focused on the cultivation of sclerotia and the direct induction of the sclerotia to fruiting; then the theory of sclerotia culture, supplementation of exogenous nutrients, conditions for fruiting, and environmental conditions were recorded in detail [20,21,22]. Based on this initial success, some commercial investors in the United States subsequently carried out factory cultivation during an initial period of time, with some scarce reports of high production; however, this seminal commercial morel cultivation was stopped around 2006 due to unstable, non-profitable production [1,23]. At that time, Segula Masaphay, an Israeli scientist, conducted some successful laboratory-scale cultivation experiments based on Ower’s technology [24,25]. Apart from that, there were no other reports on successful morel cultivation in any other countries.

The history of successful morel domestication and cultivation in China is demarcated at the beginning of the 21st century. In the early stages, the cultivation mainly focused on pure bionic cultivation, bionic cultivation by using stump trees, and mycorrhizal bionic cultivation; however, most of the studies did not have continuous follow-up [1]. The bionic cultivation of stump trees in Yunnan slightly improved some initial production and was promoted on a certain scale, but it was interrupted due to the fact that large consumption of wood violated the protection of forest resources [1,23]. From 2000–2010, the morel field cultivation in Sichuan and Chongqing gradually matured under the support of “exogenous nutrition bag” feeding technology and the discovery of “mushroom varieties feasible to produce ascomata”, which embarked on the road of commercial promotion by 2012. Based on this technology, the cultivation scale increased year by year, and simultaneously, the cultivation area spread all over China [1,26]. Parallelly, this technology has started to be introduced to France, Australia, the United States, Canada, and Turkey for trial cultivation in recent years.

With the boosting of morel field cultivation, multi-disciplinary basic biological research has gained increased attention, mainly that involving cultivation schemes, cultivation management, appropriate strains, integrative taxonomy of species suitable to be cultivated, mating type gene structure, sclerotia development, nutrient metabolism, ascomata development, and life cycle [27,28,29,30,31,32,33,34,35,36,37,38,39,40]. Studies related to “multiomics” and molecular biology technologies have also been encouraged. As a consequence, physiological, genetic, and biotechnological approaches have quickly become a new research hotspot and transformed morels into model species of macro-ascomycete cultivation [38,41,42]. Previously, there have been some reviews emphasizing the cultivation mode, nutrients and metabolism, active compounds, and life cycle [7,26,38,43,44], but less attention has been paid to aspects of paramount relevance, which have strongly affected a steady morel cultivation industry, including details of the basic principles of the cultivation technologies, the spawn technology, and characteristics related to the challenging cultivation stability. Therefore, this review will focus on the great challenges on the road to morel industry development, details of morel cultivation techniques, and potential strategies to sort out the achievements and shortcomings of basic biological research related to the maintenance of steady development of the morel industry, emphasizing the relevance of basic research areas whose knowledge needs urgent attention.

## 2. Morel Cultivation Industry

### 2.1. Evolution of Morel Cultivation

In the exploration of morel domestication and cultivation, three main cultivation technology systems have been invented: The “indoor cultivation” carried out by American scientist R.D. Ower in the 1980s [12,20,21,22], the so-called “bionic cultivation using stump trees” carried out in Yunnan, China, at the beginning of this century [45], and the “field cultivation” originated from Sichuan and Chongqing regions in the last two decades [1,23,26] (Figure 1). Although the management of the three cultivation technology systems is different, their basic nutrient metabolism principles share similarities. The idea of indoor cultivation by Ower was to cultivate sclerotia through the production process of “layered cultivation of sclerotia”, use mature sclerotia as spawn to directly sow and soak, and then induce fruiting. Thus, the emphasis of this method was placed on the cultivation of sclerotia and the direct induction of them to induce fruiting (Figure 1A). This technology was patented by Ower in the United States from 1986 to 1989 [20,21,22], and afterwards it was also adopted by some companies for morel (*M. rufobrunnea*) commercial cultivation, but it was stalled due to a lack of cultivation stability [23]. The so-called “bionic cultivation” using stump wood, invented in Yunnan, China, is a unique cultivation mode that originated from the folk. The operation process consists of stacking layers of stump wood inoculated with morel spawn and then covering them with soil. The inoculated mycelia vigorously grow, forming abundant mycelial networks that absorb the nutrients from the stump wood and, relying on natural conditions, eventually form ascomata (Figure 1B). The key point of this technique was to use stump wood of *Populus* sp. as the nutrient material in order to grow mycelia of *Morchella*, which fed on wood nutrients and eventually induced ascomata formation [45]. Despite the fact that there was some success, its weakness was that it was necessary to use great amounts of wood, which was against the protection of forest resources and therefore gradually decreased [1].

The current morel field cultivation technology, which originated in Sichuan and Chongqing, China, in the early 20th century, was gradually explored and enhanced by both scientists and amateur morel cultivation enthusiasts [1]. Its unique feature was the addition of an external feeding source, constituted by external nutrition bags. When the mycelium starts growing and, with time, establishes a mycelium network inside or on the soil surface, the external nutrition bags are placed over the abundant mycelial network, and the mycelium on the soil surface quickly colonizes the external nutrition bags. The assimilation and absorption of nutrients begin, and the nutrients in the nutrient bags are finally transferred to the mycelial network and sclerotia cells in the soil to meet the energy requirements for sexual reproduction, forming abundant ascomata (Figure 1C) [1,26].

There were two main steps in the method invented by Ower [12,20,21,22]. The first one, described frequently and mentioned above, was the production and cultivation of abundant sclerotia and the use of them as “seeds” for sowing, germination, and finally ascomata induction formation. This technology was also successfully adopted by other scientists [24,25]. The second one, mentioned but not discussed in detail by Ower, was that after sclerotia sowing, plastic bags containing rags or scraps of paper were placed on the germinating substrate, and in addition, a nutritive solution containing urea was added to the bags as nutrients to meet the needs of later fruiting formation. Due to the limited data left by Ower, these details of paramount importance have usually remained hidden from the public domain [20,21,22]. However, with the boosting of research related to the growth and development of sclerotia as a model for morel cultivation, it is possible to trace back the basic principles of the success of this technology [1,16,17,18,19,26,29]. First, studies on sclerotia formation showed that sclerotia mostly occurred in nutrient-poor environments and that sclerotia was an important lipid reservoir [16,17,18,19,29]. Second, during the cultivation process, no ascomata can be produced without the addition of external nutrient bags or nutrients added directly to the soil [36]. Additionally, it is currently known that the number of sclerotial cells and the lipid contents of the soil increase significantly with time when exogenous nutrition is added. It can be inferred that, in terms of nutrient assimilation and absorption, the process of sclerotia formation is similar to the current successful morel field cultivation. Nutrients can be stored in sclerotial cells in the soil in “rich” and “poor” areas to satisfy the appropriate amounts of energy reserve necessary for sexual reproduction in the last stage of cultivation [1]. The knowledge of these basic principles related to the differential nutrient distribution in space and their consequent energy transfer and storage in the different fungal compartments is of fundamental relevance for successful morel cultivation [1,36].

The morel nutrient demand, absorption, and storage during the different cultivation steps are different compared with other widely cultivated mushrooms, each with its own peculiarities. Although the nutrition supply methods of the three morel cultivation models are different, the basic mechanisms and dynamics involved are similar. In “Ower’s indoor” method, sclerotia are formed in the “poor” soil areas, while the nutrient “rich” organic matter layer is the energy provider, and the high-energy sclerotia structure is used to induce and stimulate ascomata formation (Figure 1A). In the “Yunnan’s stump wood outdoor” method, the stump woods are stacked in layers, or pyramids. The spawn and soil fill the gaps between the stump woods. The stump woods, including bark as a relevant source, are rich nutrient sources, and the spawn and soil inoculated between the stump wood cavities constitute “poor nutrient” areas, creating a barren nutrient condition, satisfying the assimilation and absorption nutrient requirements for morel cultivation systems (Figure 1B). The “field cultivation” method is the most advanced technique that fits the required nutrient distribution system necessary for ascomata formation. The region already colonized by the mycelial network in the soil is a nutrient “poor” area. The addition of “rich” exogenous nutrient bags allows efficient nutrient transfer from the exogenous nutrient bags to the mycelial network in the soil to achieve successful ascomata formation (Figure 1C). The “Ower’s indoor method” separates the sclerotia cultivation process (energy conversion process) from the ascomata stimulation, while the “Yunnan´s stump wood outdoor” and “field cultivation” methods integrate both stages at the right timing (Figure 1).

Currently, morel field cultivation is conducted directly on field soil as substrate, and ditching, sowing, and ridging are usually carried out in areas where the ambient temperature is lower than 20 °C (usually in autumn). Then, the exogenous nutrition bags containing wheat as the main component are sterilized, punched, and located over the mycelial beds, acting as a nutrient supplier for the mycelium network already established in the soil. After the temperature rises in the spring of the next year, mycelial growth is promoted and abundant ascomata production occurs, completing the whole cultivation cycle [1,26]. Using this technology, taking into account the local climate characteristics in different regions of China, simple flat shed sunshades (Figure 2) and a variety of basic (Figure 3) and high-tech greenhouses (Figure 4) are currently in use for the successful artificial cultivation of morels. The average yield and cultivation stability have been significantly improved by using high-quality facilities and controlled environments. A number of Chinese farms have had yields higher than 1500 g/m^2^ (Figure 3 and Figure 4).

### 2.2. Key Inventions of Successful Cultivation in China

Although Ower conducted interesting seminal technical innovations for morel indoor cultivation in the 1980s, his method was not successfully implemented worldwide. However, the field cultivation of morel in China has gradually taken shape through the efforts of many groups of scientists who currently lead this industry worldwide, mainly based on two facts.

Firstly, the discovery and application of exogenous nutrition bags. In terms of nutrient assimilation, absorption, and transportation, morel cultivation is very different compared to other cultivated edible mushrooms because it needs to create contrasting “rich” and “poor” nutrient environments through its life cycle. In fact, it was impossible to achieve these “nutrient-contrasting conditions” by using substitute materials or base materials traditionally used to stimulate other cultivated edible mushrooms. In his patent, Ower mainly focused on the method of promoting sclerotia formation and directly inducing ascomata formation from them. Despite the fact that the theory of “rich and poor nutrient contrasting conditions” was already implicit in the original indoor method of cultivation [16,17,18,19], Mr. Ower’s premature death meant that the notion of “exogenous nutrition” was obliterated for a long time [20,21,22]. The effect of exogenous nutrition bags was discovered accidentally by Chinese researchers in their long-term exploration and matured gradually through the systematic improvement of nutrition metabolism, the application of a specific methodology, and careful observation of the whole production technology [1]. This was one of the main reasons for the rapid development of the morel industry in China in the last decade.

Secondly, the discovery of cultivable and easy-fruiting species. *Morchella* is a genus with enormous diversity around the world. Globally, there are more than 300 recognized species, plus 78 phylogenetic species determined by multi-gene phylogenetic analysis [38]. At least 30 phylogenetic species are distributed in China, from which the cultivable varieties are limited to 3 to 7 species, mainly *M. importuna*, *M. sextelata,* and *M. eximia* [46]. The different cultivation potentials of morel species may be due to their differential nutritional requirements. The early failure of morel domestication may be explained in terms of an inaccurate selection of species. According to the current feeding cultivation technology, it is still impossible to achieve fruiting in all morel species, particularly in the Esculenta clade. This is a factor of paramount importance in the success of Chinese morel cultivation, which, along with an accurate and careful feeding technology, constitutes a basic element for the improving promotion of morel cultivation in China.

An additional factor that has influenced the rapid development of morel cultivation in China is its whole sociocultural agricultural system and the presence of a number of areas with favorable climate and environmental conditions. Before 2016, the cultivation areas of *Morchella* started spreading mainly in Southwest China (most conspicuously in Yunnan, Guizhou, and Sichuan provinces). The climate type in these provinces is warm and humid in the spring, and currently, it constitutes the region in which wild morels have the highest natural production. The warm and humid environmental conditions ensure the appropriate natural temperature conditions and water demand for morels to be developed from primordia to mature ascomata. In these areas, with simple shading facilities, it is possible to ensure a certain amount of morel production (Figure 2). Even in some natural mountain forests, ascomata can grow naturally after minor technological intervention. In these areas, the early morel industry in China was initially promoted, and gradually it expanded to other provinces, where different adaptations and technological enhancements boosted the industry. After 2016, China’s morel cultivation gradually spread throughout the country, especially in the Yellow River basin and the north of the country. The construction of greenhouses with higher technology in these areas, with strict regulation of temperature and humidity, allowed for the control of optimal environmental conditions, which, as a consequence, dramatically increased yields per unit area. This situation originated with the rapid development of the morel industry in Northern China in recent years (Figure 3 and Figure 4).

### 2.3. Cultivation Scale and Area in China

Since the commercialization of morel cultivation in China in 2012, the cultivation scale has increased from the initial 200 hectares to 16,466 hectares in 2021–2022. This is an awesome increase of more than 80 times during only one decade, and the cultivation area has also spread all over the country (Figure 5). The rapid growth of cultivation scale occurred in 2014–2015 and 2016–2017, with a year-on-year growth rate higher than 200%. The second boost occurred from 2017–2018, with a conspicuous increase of 97%. In addition to the evident increase in cultivation scale initiated by the morel’s cultural charm in China, and all over the world, as an edible and medicinal mushroom, there were also, in some areas, large marketing campaigns, which contributed to the success of the quick growth of this industry. However, along with this increasing growth, some challenges emerged, including, for example: (i) unstable, steady production; and (ii) great price fluctuations due to irregular production and, in some cases, insufficient market development.

The appropriate natural climate conditions in the early cultivation originated in the fact that the morel cultivation was mainly concentrated in three Chinese provinces: Sichuan, Yunnan, and Guizhou, which at the beginning of commercial production concentrated more than 75% of the total cultivated area of the country. Sichuan Province is the main cultivation area, particularly in the Chengdu Plain and nearby areas. However, after 2018, due to the cultivation failure of large areas during two consecutive years caused by the low quality of spawn, continuous harvesting without proper scientific supervision, lack of appropriate cultivation facilities, climate change, and additional unknown factors, the total cultivation area in Sichuan began to decline, with a year-on-year decrease of about 30% from 2019–2020. This large-scale failure phenomenon also occurred in Yunnan Province, and the potential explanation for these decreases might be similar to those mentioned for Sichuan Province. Outside the southwestern area, Shaanxi was the province with the fastest growth, particularly from 2021–2022. The main reason for this growth is that the warm and humid climate of Hanzhong City in Shaanxi Province is similar to that of Sichuan Province, and the low-temperature period in winter has also increased the stability of cultivation. At the same time, the disastrous failure of Chengdu Plain for two consecutive years led morel farmers to move northward. Similarly, Henan, Hebei, Shandong, and Shanxi provinces in the Yellow River basin and the North China Plain have shown a steady growth trend in recent years (Figure 5). Regional government financial support and technical guidance by enterprises are the main reasons for the conspicuous increase in cultivation areas in these provinces.

### 2.4. Challenges of Cultivation in China

Commercial morel cultivation has been maintained in China for the last 10 years. Although the cultivation scale and cultivated area continue to expand and the regional maximum yield record continued to increase, the overall steady profitability has occurred in only around 30% of the farmers, which means that around 70% of the growers have had unstable productions (with harvests per unit area lower than 150 g/m^2^), and even a proportion of these farmers have had absolute deficits. Based on long-term cultivation experience and basic biological research, we believe that there are two main reasons related to the instability of morel production. First, the appropriate scientific advice and the popularization of proper cultivation technology have not had the same expansion as morel cultivation itself. Although the cultivation technology of *Morchella* in China has been developed for more than ten years, the number of growers or technicians who have been seriously and permanently engaged for that long is less than 20% of the total. Most of them have given up after having a quick low or non-profit in their involvement in this industry. However, every year a large number of new “gold diggers” appear. As a consequence, there is a shortage of well-trained technical personnel or growers in terms of the proper selection of infrastructure conditions, understanding the relevance and management of different soil physical and chemical properties, coping with strategies to mitigate the changing climate, high-quality mycelial culture management, primordium induction technology, management and protection during the growth and development of ascomata, prevention and control of pests and diseases, and harvesting and processing technology. Secondly, with immature and non-standard spawn technology, a number of technical factors and steps must be taken to maintain high-quality standards, including, for example, the spawn quality evaluation, the standardization of proper protocols in its production, high-yield strain selection, and the management of the rapid degradation and aging of some strains. These factors have paramount importance for long-term development and steady yield in the morel cultivation technology. Additionally, a deeper understanding of some basic biological aspects, such as life cycle, mating type, asexual spore formation, sclerotia, and nutrient metabolism, will also play an important role in improving the current cultivation technology of *Morchella*. It was speculated that the failure of industrialization in the United States was a large-scale bacterial infection [23], but we consider that actually more than one single factor might have originated unstable yields and non-steady profits in the morel cultivation industry, as we have seen in different areas in China. We speculate that there is an urgent need to have a deeper understanding of the biology of cultivated morels in order to improve the challenges observed in the successful morel industry’s steady growth.

### 2.5. Economic Impact Analysis

Moving the morel cultivation process from small production to a commercial scale entails important economic issues. The main inputs involved in large-scale commercial cultivation of morels are the costs included in: (i) Production materials, which comprise spawn and exogenous nutrient bags, including wheat, sawdust, lime, gypsum, sterilization and bags production, spawn cultivation, and technical costs; with an approximate cost of USD 9500 per hectare per year; (ii) facilities, which include two options, (a) simple shading modules, including bamboo or wood to support the cultivation structures, shading nets, construction costs, and land rent; with an approximate cost of USD 3000 per hectare per year, or (b) high-tech greenhouses, including shed and sprinkler facilities (which would help to control the temperature); and land rent, with an approximate cost of USD 12,500 per hectare per year; and (iii) labor and miscellaneous expenses, including land plowing, sowing, placement of nutrient bags, daily management, and ascomata harvesting; with an approximate cost of USD 3000 per hectare per year. Therefore, the approximate costs of commercial morel production would be in the range of USD 16,000 to USD 25,000 for simple or high-tech facilities per hectare per year, correspondingly. In recent years, based on previous experiences that have shown that high-quality facilities (Figure 3 and Figure 4) produce higher profits than those in simple shading modules (Figure 2), there has been an increased trend among farmers to invest in the former. Despite successful field cultivation in China, the total production of cultivated morels is far from reaching the limits of international consumption; therefore, this factor has not yet affected international market prices. However, the two main factors that currently influence morel prices are the ascomata production instability and the harsh environmental changes, due to the fact that they may dramatically decrease morel field yields. Currently, morel production in China is very variable, ranging from 0 to 3000 g per square meter, depending on different factors, including mainly environmental conditions and economic investment in facilities.

## 3. Relevance on Biological Research for Steady Cultivation Industry

### 3.1. Taxonomy and Cultivable Germplasm Genetic Resources

The ascocarp shape of morels is strongly influenced by environmental conditions. Therefore, the classification of *Morchella* has always been a great challenge for taxonomists worldwide. However, with the advent of molecular biology, the construction of multi-genic phylogenetic analysis has been a very useful tool to establish a new criterion for the classification and identification of morels. Currently, more than 78 phylogenetic species have been identified globally, and at least 30 species have been recorded to be distributed in China [38]. Early studies on morel domestication lacked a precise species identification, so it was difficult to determine the accuracy of a robust identity of these domesticated species. It can be preliminarily stated that, except for the black species, widely cultivated at present, it is unknown if the current cultivation models might be successful for those taxa claimed to be cultivated in the past [47]. *M. rufobrunnea* was successfully cultivated indoors in America and subject to some cultivation research in lab conditions in Israel [2,22,24,25]. Du et al. (2019) pointed out that species of *M. importuna*, *M. sextelata*, *M. eximia*, *M. exuberans*, *Mel-13*, and *Mel-21* can all be domesticated [34]. However, in China, the morel species which have been cultivated at large-scale in field conditions are only *M. importuna, M. sextelata,* and *M. eximia*. We conducted a phylogenetic analysis on 185 commercially cultivated ascocarp samples collected from all over China from 2014 to 2021. The results showed that *M. sextelata* was the most abundant cultivated species, actually 155 samples were molecularly identified in this species, accounting for 83.78%; followed by 20 samples of *M. eximia*, accounting for 10.8%, and 10 samples of *M. importuna*, accounting for 5.4%. Although strains of *M. exuberans*, *Mel-13* and *Mel-21* have also been reported to produce ascomata, they have not been subject to field cultivation on a large scale. Of course, that the special nutritional supplement method of exogenous nutrition bags might not be suitable for all morel species, therefore the specific nutritional metabolism in other taxa needs further exploration.

Germplasm resources were the cornerstone of agriculture, and successful agricultural production was always based on appropriate variety breeding. At present, the morel varieties promoted in the market were mainly obtained through domestication of wild resources, and there are only a few strains obtained by cross-breeding. He et al. (2020) achieved the interspecific hybridization of *M. sextelata* and *M. importuna* strains through protoplast fusion technology, providing a potential new methodological idea for morel breeding [48]. At present, there are no new varieties of *Morchella* recognized by the Ministry of Agriculture of China. The lack of stability of commercially cultivated strains, expressed as quick aging and degradation, is definitely related to insufficient scientific knowledge of the biology of morels. As a consequence, commercially excellent varieties promoted in the market have generally gradually degenerated and been abandoned after 3–5 years of large-scale production. How to select new varieties and maintain good traits through reasonable preservation techniques is one of the big challenges to be solved in order to keep this industry alive.

There have been only a few studies related to the genetic diversity of different populations of *Morchella* in China and abroad, and even fewer studies on cultivable species. We used ITS (internal transcriber spacer) phylogenetic analysis and random amplified polymorphic markers (RAPD) to analyze 36 cultivated specimens of *Morchella* from 12 provinces in China and systematically evaluate the genetic diversity of cultivated strains. The results showed that the genetic background of cultivable varieties in China is very narrow [49]. Du et al. (2019) designed SSR (simple sequence repeats) molecular markers based on the genome data of *M. importuna* and evaluated the genetic diversity of five species of *Morchella*, pointing out that SSR had good polymorphism resolution among species and could effectively distinguish varieties [34]. The weak genetic research of germplasm morel resources has been an important factor that has restricted the breeding of new varieties.

### 3.2. Homokaryon and Heterothallic Life Cycle And Spatial Heterogeneity of Mating Type Genes

Ower et al. (1986) pointed out that mycelium germinated from single spores was able to produce ascomata [22]. Based on this record, Volk (1990) pointed out in his description that the whole life cycle could be completed through two paths: (i) Path I, or homothallic life cycle, where the mycelium produced by the germination of a single ascospore has the ability to form sclerotia and afterwards to produce ascomata; and (ii) path II, or heterothallic life cycle, where it is necessary to have two compatible mycelia, with different genotypes, germinated from two different ascospores, which must be compatible to be fused and form heterokarytic hyphae, sclerotia, and then fertile ascomata; this heterothallic life cycle has been recognized by most scientists [15]. He et al. (2017) recorded and analyzed the ascospore’s nuclear behavior of *M. importuna* and pointed out that although the ascospores of this species were multinucleate, all their nuclei originated from one of the eight nuclei formed by mitosis after early meiosis, therefore being multinucleate homokaryotic organelles [28].

The mating type gene is a gene cluster locus that controls sexual development, and it is useful to understand a fungal species’ life cycle [31,38]. The primers related to mating-type genes have been obtained and designed by shallow genome sequencing, and the monosporic populations of different species of *Morchella* have been amplified and detected by using molecular biology methods. It was preliminarily confirmed that nearly half of the phylogenetic species in the *Morchella* genus have a heterologous life cycle [27,33,50,51]. At the same time, it has been demonstrated that there are some morel species that have a homothallic life cycle [38,52]. In order to systematically explore the monosporal homokaryotic and heterothallic life cycle traits of *M. importuna*, we have sequenced, assembled, and compared the genomes of two monosporic strains with different mating type gene structures of *M. importuna*. The results showed that each monosporic strain contained only one mating type gene, Mat1-1 or Mat1-2, and the monosporic strains representing different mating types had differences at the level of genome and genes. Functional analysis of differential genes also supported the idea that the interaction between two monosporic strains was necessary to complete its life cycle [31]. The characteristics of the heterothallic life cycle of *M. importuna* were not consistent with the report that the monospore has the ability to form ascomata. In order to explore this contradiction, we further conducted some single spore and paired combination cultivation on the single spore population of the above genome-sequenced parent strain and tested and analyzed the mating type genes of ascomata (F2) and the new generation ascospores (F3) of F2 ascocarps. The results showed that although most of the single spore strains could normally produce ascomata, the F2 ascomata contained two complementary mating genotypes at the same time, i.e., compared with the parent single spore strain, there was an additional complementary mating genotype in the ascomata. The complementary genotype obtained was speculated to be caused by the spread and introduction of other fungal structures, such as mitospores. The so-called morel single spore fruiting can therefore be considered “pseudo-ascomata” [53]. Similarly, it has been shown that when cultivated with the spawn of single mating-type strains of mat1-1 or mat1-2, only one corresponding mating-type gene was detected in the mycelium and conidia. However, in contrast, in primordia, pileum and stipe were both MAT (mating types) [54].

Despite the fact that the disproportion between the two mating genotypes was found in a variety of ascomycetes, it seems to be particularly abundant in *Morchella* [55,56]. We have tested the mating type proportions of commercially cultivated strains. Tissue isolates were collected from ascomata stipe and from different parts of ascomata (including the stipe base growing in the soil, the inner and outer stipe tissues, and the inner and outer tissues of the pileus), primordia, young ascomata, and mature ascocarp, and the results showed that the two mating type genes were obviously uneven in most materials (deviation 1:1) [55]. In different parts of mature ascomata with seriously unbalanced mating types, from the base of the stipe, the middle of the stipe, the junction of the stipe and the cap, to the middle and top of the pileus, the proportion difference between the two mating type genes gradually decreased (Figure 6). Based on these results, we speculated that there was heterogeneity in the spatial distribution of the nuclei represented by the two mating genotypes and that there was competition at the nuclear level [56]. The serious disproportion was the complete loss of a mating type among the tissue isolated populations, which occurred in the tissue-isolated populations of *M. importuna*, *M. sextelata*, *M. eximia*, and Mel-21, i.e., most of the strains in these isolated populations contain only one mating karyotype [56,57]. This made it necessary to detect mating-type genes if tissue-isolated strains wanted to be used for cultivation purposes. Although the requirements for the proportion of two mating types of high-quality cultivated strains could not be given at present, at least two mating types are required to exist at the same time to produce real heterothallic fertile ascomata.

### 3.3. Mitospores and Sclerotia

There is a special kind of asexual spore named mitospores (often called conidia), which have great relevance in the life cycle of *Morchella.* The mitospores often occur in field cultivation, and they have occasionally been found in the wild [25,58], but it has been a great challenge to induce their production and germination under laboratory conditions [1,12,59,60]. Morel field cultivation showed that, 5–7 days after sowing, the mycelium growing on the surface of the soil would all be transformed into mitospores under appropriate oxygen and humidity conditions. It forms a white and powdery structure, like the frost on a winter morning, so it has frequently been called “fungal frost” [1]. The mycelium growing on the soil surface could all be transformed into mitospores, while that growing within the soil, or in areas with oxygen deficiency, would maintain its mycelial stage [60]. Usually, these mitospores would gradually disappear before ascomata formation. Cultivation experience has shown that the production of mitospores has an important impact on fruiting development. Without mitospores, fruiting would be impossible. However, the functional mechanisms involved and the biological function of mitospores are still in their infancy. We have recently conducted a detailed study on the ultrastructure and physiological characteristics of the mitospores of *Morchella.* The results showed that the cell wall of the mitospores of *M. sextelata* was very thin, which was the thinnest structure during the life cycle of *Morchella.* Based on the combination of both: (i) A very low germination rate, usually around 1/100,000; and (ii) the rapid aging of the germinating strains, it was presumed that morel mitospores had lost the rapid propagation and reproduction function like conidia and that they behaved more like microspores (spermatium) that played a mating function in fungi [60,61]. However, as a gamete, the time and space of fertilization (mating) are still unknown.

Another important structure in the life cycle of *Morchella* is the sclerotia. The sclerotia production of *Morchella* is a complex process influenced by nutrition, osmotic pressure, pH, temperature, and light, and the most significant condition has been related to the nutrient “rich” and “poor” differential conditions [16,17,18,19,30]. In addition, oxidative stress has been considered an important factor in sclerotia formation. A low concentration of hydrogen peroxide could induce the formation of sclerotia in *M. importuna* [30]. Under lab conditions, the oxidative pressure of sclerotia cells has been recorded to be lower than that of mycelial cells [30]. Gene expression related to hydrogen peroxide production has been shown to be higher in hyphal cells compared to sclerotial ones, while the expression of hydrogen peroxide-scavenging genes was higher in sclerotial cells, which, as a consequence, produce more hydrogen peroxide in the hyphal area compared to that recorded in the sclerotia [62]. Sclerotia formation is accompanied by the transformation of cell morphology and function [29,30,62]. From the vegetative mycelia to the formation and maturation of sclerotia, the cells gradually expand and twist from their initial filamentous shape into irregular spheric cells that interweave to gradually form dense sclerotia cell clusters. The cell wall gradually thickens, and the pigments become darker. Analysis of the transcriptome and subcellular structure showed that this is a process mainly involving carbohydrate, polysaccharide, hydrolase, caprolactam, β-galactosidase, disaccharide catalytic activity, and transporter activity, and finally, along with autophagy and apoptosis in sclerotia cells, which transforms the vigorous mycelia into energy storage cells (or sclerotia cells) that store great amounts of lipid substances [29,63]. The thickened cell wall structure and the storage of a large number of lipid substances not only play a role in resisting adverse environments but also are a reserve of energy for survival in either a new environment or later sexual reproduction [29,63]. It is a well-known fact that some morel species fruit abundantly in burnt sites, in particular the year following the fire event. Two possible explanations for this phenomenon might be that: (i) After forest fires there is a high release of mineral nutrients originally contained in organic form, along with a quick mineralization, which will strongly affect the normal balance of natural forest nutrient dynamics; and those abundant mineral nutrients might favor high morel ascomata formation; or (ii) the high temperatures reached during forest fires might originate quick and conspicuous changes in the original fungal community structure. If morel spores or other resistant structures, e.g., sclerotia, are able to withstand high temperatures and survive, they will become a dominant fungal group that will colonize the burnt soil substrates after fires, producing a large number of ascomata [2]. However, the precise biological mechanisms underlying the relationship between forest fires and the subsequent increase in morel ascomata in some areas are currently poorly understood.

### 3.4. Rapid Aging of Strains

Unlike most basidiomycetes, ascomycetes seem to age more easily; for example, *Podospora anserina* [64], *Neurospora crassa* [65], *Beauveria bassiana* [66], and *Metarhizium anisopliae* [67]. Hervey et al. (1978) first found that some ascospore germination of *M. esculenta* had shown growth stagnation [11]. Based on the strict subculture model, we conducted a systematic study on the aging characteristics of *M. elata*. It was found that *M. elata* had rapid aging during the subculturing process. The aging strains were also different compared to their “parental strains” and macro (sclerotia, pigment, growth rate, and colony morphology) and micro (hyphal branching, cell contents, subcellular structure) and ultrastructural aspects, suggesting that the aging of *M. elata* was a systematic, irreversible process involving autophagy and apoptosis [68]. The cultivation characteristics of the subcultured strains of *M. importuna* and *M. sextelata* showed that with aging, the degree of lipid peroxidation increased, the ability to produce sclerotia decreased, the pigment formation intensified, and the cultivation yield decreased [69]. In addition, the aging of *Morchella* was accompanied by a decrease in amylase and xylanase activities [70]. The aging process is a common phenomenon in *Morchella,* as shown in the life span of 124 strains, including *M. importuna*, *M. sextelata*, *M. eximia*, *M. elata,* and *M. crassipes* tissue-isolated strains from different parts of ascomata, different tissue block germinating strains from the same tissue part, and different monosporic strains. All tested strains would age and die in different time periods with interspecific differences [71] (Figure 7). *Morchella*’s aging directly affects the ascomata yield, making it a factor of paramount importance in the commercial morel cultivation industry. However, currently, there is a lack of research on the causes of aging and the potential successful methods to delay this aging.

### 3.5. Evaluation of High-Quality Spawn

The high-quality spawn was one of the key factors in the success of the artificial cultivation of *Morchella*. Before 2015, about 30–40% of the cultivated area in China had no production every year because non-cultivable species such as *M. angusticeps*, *M. elata,* and those belonging to the Esculenta clade were used. Currently, knowledge related to the appropriate taxa that are suitable for commercial production has largely improved. As a consequence, the phenomenon of no fruiting on a large scale due to inappropriate morel species has been greatly reduced [46,72]. The research on mating-type genes has deepened our understanding of the life cycle of heterothallism in *Morchella*. However, a large proportion of mating-type genes are still lost in *Morchella* tissue-isolated populations, which has reduced the genotypes potentially useful for industrial cultivation [38,55,57]. At the same time, as discussed above, rapid aging is a common feature in *Morchella*, and the quick aging of cultivated strains is still a factor that may sometimes cause a serious decline in yield [69]. In order to solve these challenges, it is valuable to integrate the proper molecular identification of species strains, the detection of mating type gene integrity, and the determination of strain vitality as basic quality protocols for the evaluation of morel spawn to be used in commercial production [72]. Using this IMV (identity, mating type, and vitality) evaluation system, we evaluated and tested about 2400 hectares of source spawn in the last three years, and the results showed that the cultivation stability had improved significantly, which could achieve a consistent and stable yield on a large scale, on multiple farms, and in consecutive years. Of course, we have to recognize that currently, there is only the evaluation of the growth rate, hyphal branching, sclerotia, pigment, life span, peroxidase, amylase, and xylanase of the spawn, and there is an urgent need to increase the knowledge related to additional specific and direct indicators that might show high-quality strains with high production and appropriate biochemical traits. In summary, it is mandatory that before inoculation, it is necessary to determine the identity and genotype integrity of the strain to be used in commercial production and, at the same time, ensure the high vitality of the spawn, which is crucial for stable and high-yield morel production.

### 3.6. Diseases and Pests

The commercial production of morels, as with any agricultural crop, is vulnerable to the invasion of diseases and pests. These include insect pests and bacterial and fungal diseases. However, the scientific knowledge related to this topic is still in its infancy. Morel fungus disease is characterized by evident white villous lesions. The affected part withers and then causes the ascomata to rot and deform, and additionally, if not controlled appropriately, it is highly infectious [73]. It is very easy to cause large-scale outbreaks, particularly in high-temperature and high-humidity environments. Currently, only physical methods could be used for prevention and control, such as reducing the environmental humidity and temperature to decrease the spread and reproduction of the involved pathogens or removing the diseased ascomata (with additional physical isolation and proper disposal). Morel fungus diseases might account for around 25% of economic losses every year [73]. *Fusarium incarnatum*—*F. equiseti* [74], *Paecilomyces penicillatus* [75], *Diploöspora longispora* [76], *Cladobotryum protrusum* [77], and *Lecanicillium aphanocladii* [78]—have been reported as the potential causal pathogenic agents of the ascomata fungal diseases. However, our epidemic analysis in 32 farms in 18 provinces of major cultivation scale in China showed that *D. longispora* is the real culprit since 93.75% of the fungal diseased ascomata showed that this fungal pathogen presented the highest sequence abundance, followed by *Clonostachys solani* with only 5.04% [73]. The traceability analysis of *D. longispora* indicated that it widely existed in soil and air, was enriched by the nutrition or environmental conditions in the morel cultivation systems, and was gradually transferred to the ascomata at the fruiting stage. When high temperatures and high humidity occurred, it turned into a pathogenic fungus invading the ascomata [73]. Tan (2021) pointed out that when *Paecilomyces* (which should be *Diploöspora*) in the soil matrix grows in large abundance, it could even cause the danger of non-fruiting. In fact, this pathogenic fungus was also the main cause of the serious decline in yield in continuous cropping. The continuous cropping caused an imbalance of microbial populations in the soil matrix, especially by enriching harmful microorganisms (including bacteria and fungi), which harmed the growth of morel and the development of ascomata [39]. Wang et al. (2021) pointed out that the inhibition of the mycelial growth of *M. sextelata* was caused by a soluble compound rather than volatile compounds secreted by *Paecilomyces penicillatus*, based on confrontation tests between *P. penicillatus* and *M. sextelata*, and analyzed the characteristics of the *P. penicillatus* fungal cell wall-degrading enzyme system and the antifungal, antibacterial, and insecticidal activity of secondary metabolite gene clusters at the genomic level [79]. 1-octen-3-ol is a volatile compound of mushrooms with broad-spectrum antifungal activity. Compared to *P. penicillatus*, *M. sextelata* has a higher tolerance concentration to 1-octen-3-ol, and this compound can significantly change the microbial community structure of the whole growing cycle of *M. sextelata* and has a good effect on inhibiting the outbreak of *P. penicillatus* disease and therefore increasing *M. sextelata* yields, which provides an encouraging strategy for control of morel fungal disease [80].

### 3.7. Omics and Genetic Engineering Technology

Omics is a discipline that studies the structure, composition, interaction, function, and evolution of genes, proteins, or metabolites from a complementary perspective. Compared with other crops, *Morchella* has been studied relatively recently; however, it has attracted great attention due to its industrial importance. Up to now, high-quality genomes of *Morchella* have been published for *M. importuna* [31], *M. sextelata* [35], and *M. crassipes* [51], as well as Chai et al. (2017), Murat et al. (2018), Tan et al. (2019), and Du et al. (2020), which sequenced the shallow genome of *M. eximia*, *Mes-21*. This information would facilitate the exploration of specific genes, phylogeny, metabolism, and species evolution in *Morchella* [36,38,50,81,82].

Multiomics studies related to *Morchella* have been a prominent trend in recent years. The research subjects have mainly focused on the growth and development of *M. importuna* and *M. sextelata*, including the comparative transcriptome, metabolome, secretome, and proteome at the stage of the growth and development of mycelia, sclerotia, and ascomata [32,36,40,62,63,83,84,85,86,87,88,89,90]. However, these studies have mostly concentrated at the upper level, and the exploration of functional genes, regulatory networks, or metabolic pathways that respond to development needs to be deepened. Amplicon sequencing of microbial communities has also been used to reveal the development of morel and its interaction with the environment [32,36,39,73,85,87,91,92]. However, the microbiomes that influence or trigger ascomata formation have received little attention. Despite some seminal studies showing that there is a large diversity of prokaryotic and eukaryotic microorganisms that inhabit the morel ascomata, which might affect their growth [32,73,93].

The study of functional genes has been inseparable from the gene editing system. Although it has gained attention recently, the genetic transformation system of *Morchella* has been preliminarily established. Lv et al. (2018) first reported the construction of a genetic transformation system for *M. importuna* through an *Agrobacterium*-mediated method [41], and then the research team added nuclear localization tags to eGFP and mCherry fluorescence, respectively, with the help of this system, thus realizing the transformation of two different mating strains of *M. importuna*. The pairing of transformants showed that they can be well fused, which intuitively displayed the heterothallism of *M. importuna* [42]. The successful construction of a genetic transformation system would provide additional technical support for further study on gene function, regulation, and development in *Morchella*.

## 4. Conclusions

Although the domestication and cultivation of *Morchella* have been achieved with the efforts of a large number of scientific researchers and farmers, there is still a big gap between morel cultivation and the domestication of other traditional edible mushrooms. *Morchella* presents distinctive characteristics, including genetic instability, rapid aging, high sensitivity to environmental conditions, heterothallism, and easy mating genotype loss. Therefore, it is urgent to develop and formulate high-standard protocols for germplasm evaluation and technical specifications for high-quality spawn. Additionally, there is an urgent need to strengthen basic scientific research in areas of paramount importance in order to ensure steady commercial production. These should include: (i) Systematic analyses of the morel genetics and development of more stable SSR, SNP (single nucleotide polymorphism) or Inder markers for native germplasm by using omics research; (ii) domestication of wild germplasm by using beneficial mutation-, crossed-, and genetic engineering- breeding; (iii) research at different complexity scales involving cells, nucleic acids, and/or genes related to the exploration of causes of genetic instability; (iv) Systematic studies of internal and external factors that induce the rapid aging of *Morchella* strains, analyzing the intrinsic mechanism, and delaying aging or constructing anti-aging strategies through chemical or genetic engineering techniques; and (v) in-depth studies on nutrition metabolism, nutrient flow, gene and metabolic regulation, and interaction with other microorganisms. At the same time, the imbalance of allelopathy, nutrients, and functional microbial communities will contribute to an understanding of the thorny “continuous cropping” problem of morel cultivation. The study of the interaction mechanisms of *Morchella* between biological and abiotic factors can provide ideas for boosting the morel’s indoor industrial cultivation in the near future. The inherent characteristics of *Morchella*, such as its short growth cycle, mushroom production at medium and low temperatures, and high market price, are great incentives to invest in the topics mentioned above, which would quickly develop a stable and profitable industry. Learning from the experiences of established industrial cultivation of conventional basidiomycete edible fungi might also speed up the research on the automated cultivation of *Morchella.*

In general, currently, morel cultivation is a profitable business with great potential to grow at an international level. Despite currently covering around 16,500 ha in China, this morel field cultivation area is still small in the country, not to mention at the global level, where there is currently an open and profitable business opportunity. One of the main factors is the high economic value of dried morels, which on average ranges around USD 200 per kilogram in the European and North American markets. However, its industrial growth faces big challenges, the main one being to keep production steady, which urgently needs to promote a deep understanding of the scientific basic aspects mentioned before. Global boosting of morel cultivation would play a role in meeting different United Nations Global Sustainable Development Goals (generally known as GSDG) including: (i) contribution to food security by influencing the production of nutritious foods with health-promoting benefits; (ii) preservation of a group of genetic resources which currently at risk, even before they are known, due to high deforestation rates; (iii) economic development; (iv) industrial innovation; and (v) partnerships for the goals, because its boosting will only be possible by developing strategic alliances involving policy makers, entrepreneurs, scientists, and different social sectors, placing emphasis on rural population development.

## Figures and Tables

**Figure 1 jof-09-00855-f001:**
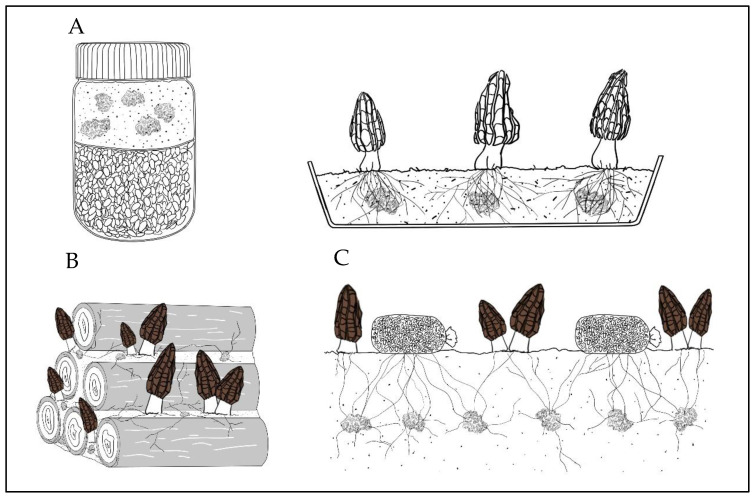
Schematic diagram of the three successful cultivation technologies of morels: (**A**) Ower’s indoor method, using sclerotia as “seeds”, sown in shallow trays; (**B**) Yunnan’s stump wood outdoor method, using the addition of inoculated mycelia and soil among the cavities of *Populus* sp. stumps, which promotes mycelial growth by nutrient absorption from bark, sclerotia formation, and finally achieves ascomata development; and (**C**) field cultivation method, where the addition of external nutrition bags boosts the ascomata formation.

**Figure 2 jof-09-00855-f002:**
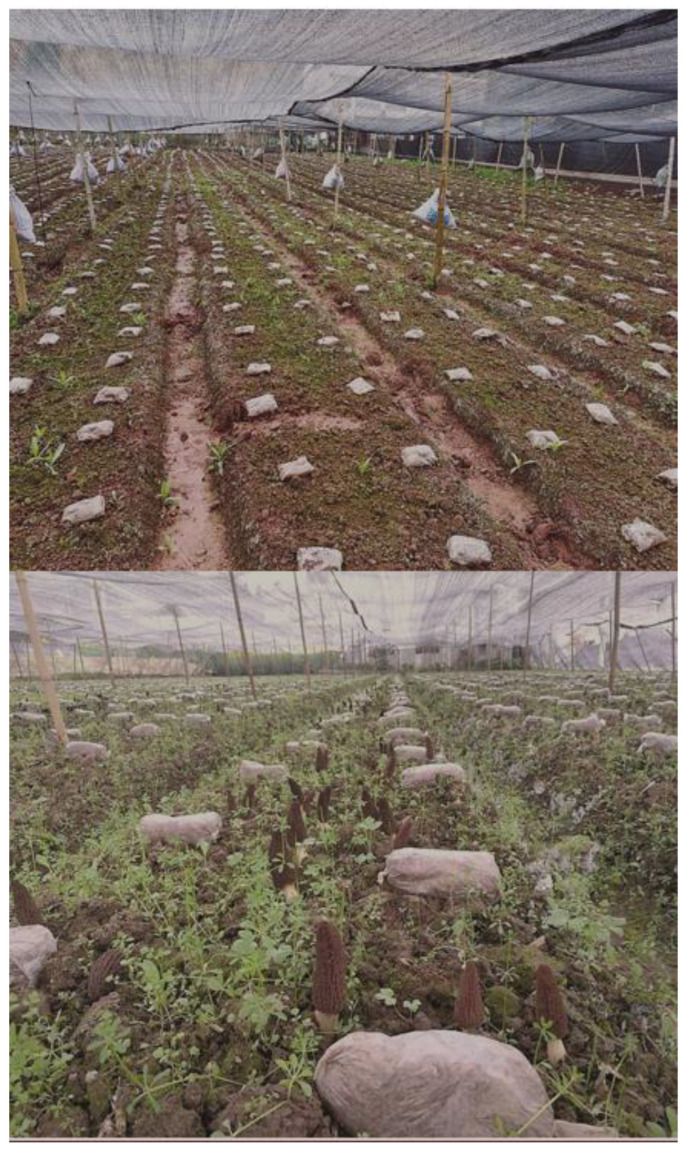
Morel cultivation facilities with small financial investment using flat-shed modules, which are usual in Southwest China and only help with low control of environmental natural weather outdoor conditions. These basic facilities might have no production (above) or low yields, for example, for *M. importuna* (below). Usually, these elementary facilities are mostly used in areas where weather conditions are not extreme.

**Figure 3 jof-09-00855-f003:**
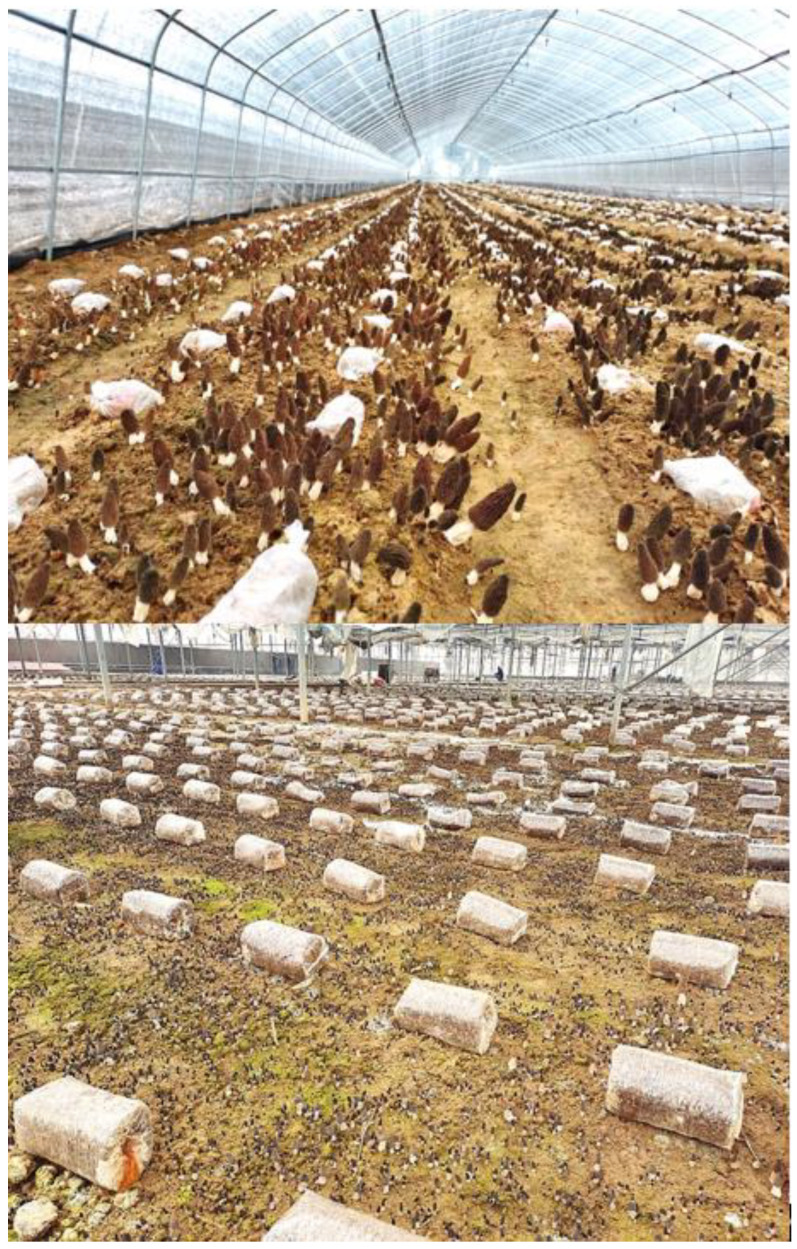
*Morchella sextelata* cultivated in facilities with moderate financial investment using conventional vegetable greenhouses with plastic (above) or glass (below) sheeting, which allows control of temperate outdoor weather fluctuations. Plastic vegetable greenhouses are the most commonly used, and high yields are clearly observed when using these types of greenhouses.

**Figure 4 jof-09-00855-f004:**
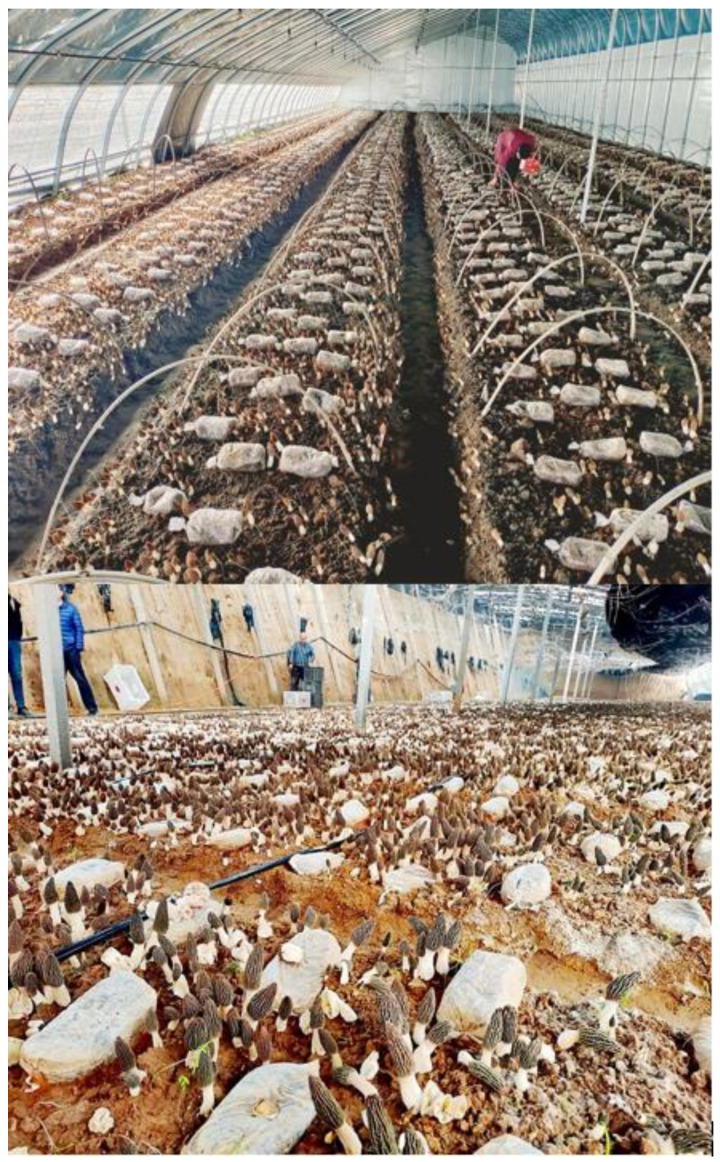
*Morchella sextelata* cultivated in facilities with high investment using automated temperature and humidity control greenhouses. Very high yields are clearly observed in both greenhouses (above and below). These facilities are appropriate for any kind of weather condition, including areas with harsh or changing outdoor climate conditions. Cotton felt was added to the plastic greenhouse sheeting for better temperature control (above); and the same material plus removal of 1 m-deep soil was used in order to increase the insulation and temperature control (below).

**Figure 5 jof-09-00855-f005:**
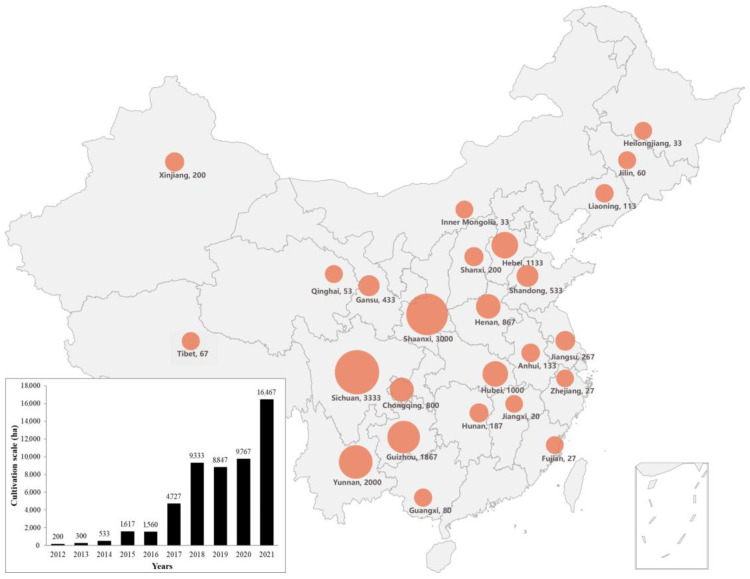
Growth and distribution of morel industry in China. The graph shows the annual change trend of morel cultivated area in China from 2012 to 2021, and the map shows the current areas in the country’s provinces from 2021 to 2022, expressed in hectares after the province names and correlated with the size of the salmon-colored circles for each province.

**Figure 6 jof-09-00855-f006:**
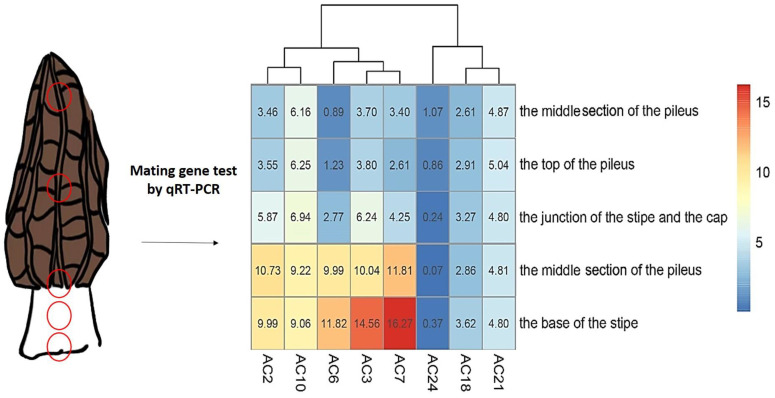
Heat map of two mating types in five compartments of eight *Morchella importuna* samples. qRT-PCR (quantitative real-time polymerase chain reaction) detection of gene contents of two mating types (*Mat1-1-1* and *Mat1-2-1*) showed significant differences in absolute values of CT (cycle threshold) in the different compartments (shown on the left side as red circles). AC series represents samples from eight different ascomata, and heatmap data are from He et al. [55].

**Figure 7 jof-09-00855-f007:**
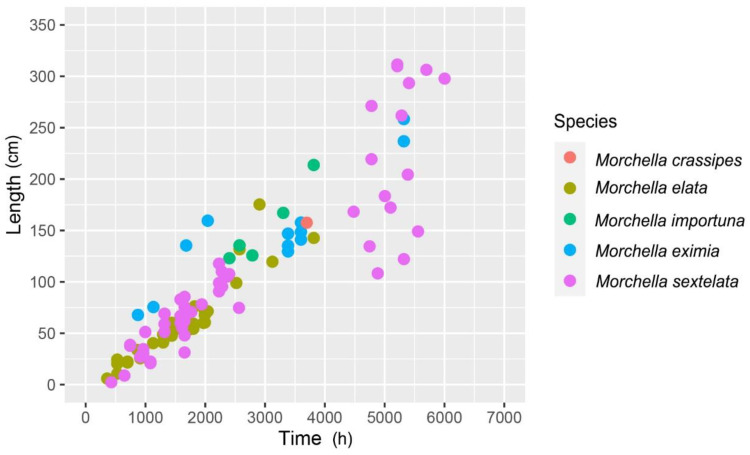
The lifespan (time) and growth distance (length) of the different *Morchella* under subculture.

## Data Availability

Not applicable.

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
