# Peer review of "Large-Scale Field Cultivation of Morchella and Relevance of Basic Knowledge for Its Steady Production"

_jof, 2023, doi:10.3390/jof9080855_

Round 1

Reviewer 1 Report

The authors provide a very up-to-date review of the industrial cultivation of Morchella, with an in-depth study of the techniques used and analyzing the key role that Chinese technology has played in large-scale development.

Unlike other reviews already written on this subject, the originality of this manuscript lies in the fact that it analyses the growth, evolution, and distribution of industry in China and the climatic, human, and technological causes by which farmers have modified their production.

The review is very well written, clear, and orderly, which makes it easy to read. In addition, the authors are very knowledgeable about the subject, providing their own data and some new, hitherto unpublished data.

There are some factors to take into consideration:

- One well-known observation about the ecology of morels is that they fruit abundantly in burnt sites, in particular the year following the fire event, could the authors relate this fact to the technological systems used in the cultivation of morels? 

- Since Morchella sextelata is one of the most cultivated species in China, it would be interesting to include in the review the article by Liu et al. (2022), concerning mating-type genes.

Liu, Q.; Qu, S.; He, G.; Wei, J.; Dong, C. Mating-Type Genes Play an Important Role in Fruiting Body Development in Morchella sextelata. J. Fungi 2022, 8, 564.

- Figures 2, 3 and 4 illustrate incredible productions of morels, if possible, could the authors indicate which Morchella species are shown in these pictures?

- Line 248, correct the year 2106

- Lines 289-290: correct editing error

Reviewer 2 Report

This manuscript presents a review on large-scale commercial cultivation of Morchella and relevance of basic biological knowledge to maintain steady industrial production. The content of the paper is strongly related to the scope of the journal but it has to be improved considering the below comments.

1.       Please reformulate the title because it is too lengthy.

2.       Some numerical data regarding the production rates on global scale could be stated in the introduction section to point out the importance of this mushroom. Also, some data on bioactive compounds could be mentioned in the introduction section.

3.       Considering that the authors reflect on the large-scale commercial cultivation of Morchella mushrooms it would be very important to present some economic data. What are the financial involvements of a large scale production? How can be the process scaled and what is the relationship between scaling the production and economics. What are the equipment and O&M costs? Sensitivity analysis also could be of interest…How market price impacts the large-scale commercial cultivation of Morchella.

4.       Environmental impact of large-scale commercial cultivation of Morchella should be also incorporated in the review. How different stages associated with the large-scale commercial cultivation of Morchella mushrooms impact biodiversity, carbon footprint, etc.?

5.       Please add some numerical data to the conclusion section and define some future perspectives that could arise from the literature overview.

6.       The authors could define a list of abbreviations.

Minor editing of English language required

Round 2

Reviewer 2 Report

The article may be published in its current form.